# Anatomical and Digital Image Analysis of Flavonoid-Mediated Grain Coloration in Rye (*Secale cereale* L.)

**DOI:** 10.3390/plants14162557

**Published:** 2025-08-17

**Authors:** Pavel A. Zykin, Elena A. Andreeva, Natalia V. Tsvetkova, Andrey N. Bulanov, Anatoly V. Voylokov

**Affiliations:** 1Faculty of Biology, St. Petersburg State University, Universiteskaya nab. 7/9, St. Petersburg 199034, Russia; pavel.zykin@spbu.ru (P.A.Z.); n.tswetkowa@spbu.ru (N.V.T.); 2Laboratory of Plant Genetics and Biotechnology, Vavilov Institute of General Genetics Russian Academy of Sciences, Gubkina st. 3, Moscow 119991, Russia; bulanov@vigg.ru (A.N.B.); voylokov@vigg.ru (A.V.V.)

**Keywords:** rye, image analysis, grain color, anthocyanins, proanthocyanidins, MALDI-IMS, CIELAB

## Abstract

Rye exhibits high diversity in grain coloration among small cereals, which is mainly linked to the presence of colored flavonoids synthesized in the outer layers of the kernel. This variability is not yet sufficiently described from colorimetric, cytological, and biochemical points of view. In this study, the localization of flavonoid pigments, anthocyanins and proanthocyanidins (PAs), was analyzed across different grain tissues in 26 rye lines with identified anthocyanin grain color genes. Grain coloration was objectively characterized using the CIELAB color coordinates and the GrainScan software for image analysis of individual grains. The distribution of anthocyanins and PAs was investigated through light microscopy and matrix-assisted laser desorption/ionization imaging mass spectrometry (MALDI-IMS) on longitudinal and cross sections of the grains. The results revealed that violet-grained lines accumulate anthocyanins in the pericarp, while green-grained lines contain anthocyanins in the aleurone layer. MALDI-IMS confirmed the presence of specific anthocyanins: cyanidins in the pericarp of violet-grained lines and delphinidins in the aleurone layer of green-grained lines. All studied lines, except for the anthocyanin-less vi3 mutants, exhibited PAs in the brown-colored testa. Four main color groups of the rye grains (yellow, green, brown, and violet) could be clearly differentiated using the CIE color coordinate h°.

## 1. Introduction

The health benefits of grain-based nutrition are presumed to be enhanced by using products made from colored grains compared to uncolored (yellow) grains [1]. In cereal crops such as maize, wheat, rice, and barley, these benefits are primarily attributed to the presence of colored flavonoids. The overall grain color in cereals is determined by pigments synthesized in both maternal tissues (fruit and seed coats) and syngamic tissues (aleurone layer and starchy endosperm). Blue-purple anthocyanins, red-brown proanthocyanidins (PAs, also known as condensed tannins), and phlobaphenes accumulate in the outer layers of grains, alongside yellow glycosides of flavones, flavonols, and flavanols. Carotenoids, on the other hand, are responsible for the yellow coloration of the endosperm. Additional factors, such as colorless co-pigments, complexes of pigments and co-pigments with metal ions, tissue-specific pH levels, and the structural properties of the grain surface and endosperm can influence the perception of grain color by the human eye [2].

Among cereal crops, rye (*Secale cereale* L.) exhibits remarkable diversity in grain coloration (Figure 1). Studies of numerous rye accessions [3] have classified rye grains into a wide range of colors, including green, yellow, white, purple, brown, black, and transitional shades such as yellow-green, light green, gray-green, dark green, yellow-brown, and light brown. Open-pollinated rye varieties are polymorphic for grain color and have grains of most of the described colors, while purple and black grains are rare and are typically found only in populations of weedy rye, as well as in lines derived from their hybridization with cultivated rye. Despite this diversity, the literature lacks detailed descriptions of the pigments present in the tissues of rye caryopses of different colors. However, anatomical study of yellow, green, and brown grains have provided some insights into the coloration of grain layers [4].

The study [3], conducted on open-pollinated Russian rye varieties polymorphic for grain color, revealed that in yellow and green grains, all cell layers of the fruit coat are yellow, with the upper two layers being more intensely colored and less transparent. In brown grains, these layers are brown and also more intensely colored than the underlying layers. Additionally, grains of all three colors possess a brown-colored seed coat (testa), while the aleurone layer appears as blue-green transparent flakes, with the highest color intensity observed in green grains.

Visual descriptions of grain coats have led to a speculative hypothesis about the diversity of rye grain coloration. In the monograph Rye, V. D. Kobylyansky [3] presented a table summarizing the color diversity of rye grains as a combination of color gradations in the aleurone (colorless to blue), testa (colorless to red-brown to dark brown), and pericarp (colorless to yellow to purple/violet). However, no data is available on the specific distribution of pigments in the tissues of rye grains with different colors. The Peterhof genetic collection of rye [5,6,7] includes lines with stable violet, brown, green, and yellow grain coloration. The genetic basis of anthocyanin pigmentation in these lines [5,8] and the composition of anthocyanins in their grains have been previously characterized [9].

The aim of this study is to investigate the localization of pigments in the tissues of rye grains from lines with different grain colors. We employed light microscopy and matrix-assisted laser desorption/ionization imaging mass spectrometry (MALDI-IMS) to study the distribution of anthocyanins and PAs. To provide an objective, numerical description of grain coloration, we used the GrainScan software [10] with a color-calibrated flatbed scanner for fast and low-cost analysis of grain size and color measurements. Color is described in a device-independent CIELAB color space, which includes all perceivable colors and represents them through the values of three coordinates (L*, a*, b*). The results of our study can be applied to develop fast and non-destructive optical methods for grain sorting, to differentiate breeding material and associate color coordinates with the composition of colored flavonoids or other pigments having health-improving effects under human consumption of rye grain-based products.

## 2. Results

### 2.1. Color Coordinates Analysis

The results of the digital evaluation of grain color are presented in Table 1. The 26 lines were divided into five groups based on the presence of some anthocyanin pigmentation genes, resulting in different grain and vegetative organ colors. These groups, consistent with a previous publication [5], were classified as follows: anthocyanin-less (lines 1–7), yellow-grained with anthocyanin pigmentation in vegetative tissues (lines 8–11), brown-grained (lines 12–15), green-grained (lines 16–22), and violet-grained (lines 23–26). The data was analyzed by comparing the means of the five groups using analysis of variances (ANOVA) and Tukey’s post-hoc test at *p* ≤ 0.05 [11].

L* (lightness). The anthocyanin-less, yellow-grained, and green-grained lines showed no significant differences in L* values, with means of 48.83, 45.74, and 47.11, respectively. In contrast, the brown-grained lines had a lower L* value of 40.00, while the violet-grained lines, as expected, exhibited the lowest L* value of 28.70.

a* (greenness-redness). The green-grained lines had the lowest a* value (2.14), indicating a greener hue. The anthocyanin-less and yellow-grained lines displayed intermediate and nearly equal a* values of 3.53 and 3.21, respectively. The brown-grained and violet-grained lines had the highest a* values, which were almost identical at 4.66 and 4.67, respectively. This trend reflects a gradual reddening of grain color, progressing from green through yellow to brown and violet.

b* (blueness-yellowness). The violet-grained lines had the lowest b* value (10.68), while the green-grained lines exhibited an intermediate value of 13.79. The anthocyanin-less lines had the highest b* value (15.73). The yellow- and brown-grained lines, with b* values of 14.36 and 15.16, respectively, did not differ significantly from the anthocyanin-less and green-grained lines. Interestingly, this parameter appears to correlate closely with the presence of specific anthocyanins: cyanidin derivatives in the pericarp of violet-grained lines, delphinidin derivatives in the aleurone of green-grained lines, and the absence of these pigments in anthocyanin-less lines (see Section 2.2. “Anthocyanin and PAs localization in rye kernel”).

C* (saturation or color purity). For this coordinate, only the anthocyanin-less lines (16.13) and violet-grained lines (11.71) differed significantly from each other.

h° (hue, actual color). The h° coordinate best reflects the visual classification of grains by color. The yellow grains of the anthocyanin-less lines (77.36) did not differ significantly from those of the yellow-grained lines (77.45). However, the brown-grained (72.81), green-grained (81.30), and violet-grained (66.61) lines differed significantly from the yellow-grained lines and from each other. This pattern likely reflects the presence of distinct pigments: as yet unidentified pigments in the pericarp of brown grains, delphinidin derivatives in the aleurone of green grains, and cyanidin and peonidin derivatives in the pericarp of violet grains. In contrast, yellow grains lack these pigments. Notably, the violet-grained lines exhibited the greatest intra-group variability in both directly measured (L*, a*, b*) and calculated (C*, h°) parameters. This variability may correspond to the previously observed diversity in anthocyanin composition in these lines, as determined by high-performance liquid chromatography mass spectrometry (HPLC-MS) [9].

### 2.2. Anthocyanin and PAs Localization in Rye Kernel

Histological slices of yellow, brown, green, and violet kernels were examined under a microscope to characterize the original coloration of their upper layers. The violet pigment is concentrated in the pericarp of violet grains, while the blue pigment is localized in the aleurone layer of grains that appear green on the surface (Figure 2). Additionally, a brownish, yet unidentified pigment, was observed in the pericarp of brown grains. The majority of the studied lines, including yellow-grained ones, possess a naturally brown testa. However, the intensity of this brown coloration varies significantly.

Vanillin-HCl staining revealed the presence of PAs in the testa (Figure 3C). However, the color of the stained product was similar to that of residual anthocyanins, requiring a comparison between vanillin-HCl-treated (Figure 3C) and untreated (Figure 3A) slices. Furthermore, the vanillin-HCl stain of rye is short-lived, lasting only a few hours, which necessitates immediate microscopic analysis.

In contrast, DMACA (4-dimethylaminocinnamaldehyde) staining (Figure 3D,E) also localized PAs in the testa but produced a blue-colored product, allowing it to be easily distinguished from the red-colored residual anthocyanins on the same slice (Figure 3E). Additionally, the DMACA stain is more stable, lasting for a considerable period, which enables the automated capture of multiple fields of view for high-resolution panoramic image montage, allowing local changes in different parts of the grain to be noticed.

The application of hydrophobic 1-hexadecene as a mounting medium for fresh frozen-cut grains effectively prevents the delocalization of water-soluble components, provided that microscopy is performed shortly after mounting. All three grain coats—aleurone, testa, and pericarp (Figure 3A,B)—were clearly differentiated in both violet (Figure 4A) and green (Figure 4B) fresh frozen-cut grains without significant water-soluble anthocyanin delocalization.

Violet-grained lines (Figure 2 and Figure 4A,C,E) contain anthocyanins primarily in the pericarp, with peonidin- and cyanidin-rutinoside being the main compounds, as previously identified by HPLC-MS [9]. These lines also exhibit a brown-colored testa and a colorless aleurone layer. The localization of peonidin-rutinoside in the pericarp was confirmed by matrix-assisted laser desorption/ionization imaging mass spectrometry (MALDI-IMS) (Figure 5E). The presence of the molecular ion (*m*/*z* 609.18), along with tandem mass spectrometry (MS/MS) fragments (*m*/*z* 463.12) and the aglycone fragment (*m*/*z* 301.07), provided further evidence for this localization. The change in ion intensities (Figure 5A–C,E), higher near the embryo and lower at the opposite side of the kernel, corresponds to a gradient in pigment distribution as revealed by histological analysis (Figure 4E).

The green-grained line L8 (Table 2, Figure 2 and Figure 4B,D) features a brown-colored testa and a light blue-colored aleurone layer. The primary anthocyanin in this line, as identified by HPLC-MS [9], is delphinidin 3-*O*-rutinoside. MALDI-IMS confirmed the localization of this compound in the aleurone layer, as indicated by the presence of its molecular ion (*m*/*z* 611.16) (Figure 5F). The change in ion intensities is more pronounced, with most of the signal being near the embryo (Figure 5F), closely following the pigment distribution pattern in histological analysis (Figure 4B).

DMACA staining revealed the presence of PAs in the majority of the studied lines. Among the anthocyanin-less lines, variability in staining intensity was observed: line vi3 showed a complete absence of staining, line vi6 exhibited weak staining, line vi1 displayed spotted staining, and lines vi2, vi4, and vi5 showed prominent staining (Figure 6). Notably, the blue coloration and corresponding PAs concentration was more abundant near the embryo and at the bottom of the crease.

## 3. Discussion

### 3.1. Color Coordinates

CIELAB color coordinates are widely used to objectively describe and classify biological objects, including plants, and to develop indirect methods for pigment analysis [12,13]. For example, in colored varieties of grape (*Vitis vinifera* L.), all color coordinates (L*, a*, b*, and C*) exhibit a negative correlation with total anthocyanin content (TAC), while h° shows a positive correlation [14]. Similar correlations between individual anthocyanins and color coordinates have been observed in some cereals [15], where grain or flour color parameters were linked to pigment content. In black rice (*Oryza sativa* L.), the content of anthocyanins (cyanidin 3-*O*-glucoside and peonidin 3-*O*-glucoside) in flour was negatively correlated with L*, b*, C*, and h° values, while PAs in red rice flour showed a positive correlation with a* [15,16].

The results of grain colorimetry in bread wheat (*Triticum aestivum* L.) [17] align with our findings in rye. In wheat, the presence of blue pigments in the aleurone, purple pigments in the pericarp, and red pigments in the testa has been confirmed [17], consistent with previously published data. The black (dark purple) color of wheat grains is attributed to the presence of anthocyanins in both the aleurone and pericarp. The color coordinates of wheat varieties with white (yellow), blue, and purple grains overlap with those of rye grains exhibiting yellow, green, and violet coloration, respectively. For instance, the average h° values for these groups in wheat are 73.0, 81.4, and 61.7, while in rye, they are 77.45, 81.3, and 66.61, respectively. Since CIELAB color space is device-independent [12], these values can be directly compared.

Based on previous studies of rye [9], wheat [17], and grape [13], we observed similar relationships between anthocyanin concentration and color coordinates (Table 1). Specifically, L*, b*, and C* displayed negative correlations with total anthocyanin content (TAC). However, unlike the findings in [9,13,17], h° also exhibited a negative correlation with TAC in our study. Additionally, the a* coordinate for lines with the highest TAC (violet-colored lines) was similar to that of brown-colored lines lacking anthocyanins in grains [9], indicating that a* is not a reliable indicator for assessing anthocyanin content in rye. The minimum value of a*, combined with the maximum value of h°, was observed in green-grained rye and blue-grained wheat [17]. In violet-grained rye, the minimum values of L*, b*, C*, and h° were characteristic, whereas in purple-grained wheat, this pattern was observed only for L* and h°. These similarities may be attributed to the predominance of delphinidin derivatives in the aleurone and cyanidin derivatives in the pericarp in both cereal species. However, some discrepancies may arise from the presence of additional pigments or co-pigments, morphological differences, or environmental and statistical variations.

It is well established that both the chemical composition and structure of individual anthocyanins influence color coordinates [18]. The significant variation in all coordinates observed in violet-grained rye lines (Table 1) may reflect differences in anthocyanin content and composition among these lines [9]. This variability opens up prospects for developing a non-destructive method to determine anthocyanin content in rye lines and hybrids with violet grains. Additionally, the inclusion of individual grain size parameters provided by GrainScan expands the analytical capabilities of such methods.

An attempt was made [19] to associate grain color with the content of flavones and cinnamic acid derivatives, which are effective co-pigments, in 12 open-pollinated rye varieties. These varieties were divided into four groups based on the visual assessment of grain color in bulk: yellow/tan, blue/gray, blue/green, and green. The studied populations exhibited significant variation in the content of these compounds, with ranges of 57–137 µg/g for flavones and 9.2–93 µg/g for cinnamic acid derivatives [19]. However, open-pollinated rye varieties are typically polymorphic for grain color, and the color classification was based on an “average” assessment for each variety. It is possible that the lack of individual grain color classification prevented the establishment of a correlation between grain color and co-pigment content. Additionally, differences in co-pigment content may only be reflected in grain color when anthocyanins and co-pigments are co-localized in the same cellular compartment. In this case, co-localization could only occur in the vacuoles of endosperm cells, as the studied varieties lacked anthocyanins in the pericarp.

Our results suggest that while the development of a small research-scale grain sorting machine based on rye grain color is feasible, several issues must be addressed for better performance. Due to the small differences in C* and h° coordinates between groups, the recommended bit depth for the sensor should be at least 12 bits per channel. Special attention should be given to lighting conditions (e.g., using a D65 illuminant), ensuring even illumination intensity. The uneven coloration of grains, with higher pigment concentration near the embryo and crease, necessitates uniform grain orientation during imaging. Additionally, surface properties such as smoothness, waxiness, and grain humidity can affect surface reflectance. To minimize these effects, the angle between the illumination source and the camera should be carefully optimized. Alternatively, polarized illumination can be used to reduce reflectance artifacts. Given the moderate intra-group variability, which can be influenced by factors such as grain orientation, reflectance, and size, an imaging detector (e.g., a camera) is preferred. Most of these factors can be accounted for by analyzing images of moderate resolution. Furthermore, our classification model can be enhanced by incorporating machine-learning approaches, rather than relying solely on threshold-based methods, to better recognize subtle differences in grain color.

### 3.2. Pigment Localization

Tissue-specific localization of colored flavonoids has been well-documented in the caryopses of cereals such as maize (*Zea mays* L.), rice (*Oryza sativa* L.), sorghum (*Sorghum bicolor* L.), wheat (*Triticum aestivum* L.), and barley (*Hordeum vulgare* L.) [20]. The observed inter-species differences in flavonoid composition and localization likely arose during the evolution of these crops through processes such as adaptation, domestication, and breeding. Duplications and functional divergence of flavonoid biosynthetic genes and regulatory genes, particularly those comprising the R2R3-MYB–bHLH–WDR (MBW) transcription factor complex, may have led to the emergence of specific branches of flavonoid biosynthesis and unique regulatory features [20].

Despite the existence of wild ancestral forms and local varieties with colored grains, the majority of cultivated cereals are represented by varieties with uncolored grains. This is largely due to long-term breeding efforts aimed at improving technological and taste qualities, which are often negatively associated with grain color [21]. However, recent breeding efforts have shifted toward developing cereal varieties with high levels of colored flavonoids, which correlate with antioxidant capacity and other health-promoting effects of whole-grain products [1].

Colored flavonoids in cereals are synthesized in the aleurone, testa, and pericarp of the caryopsis, with specific regulation in each tissue by corresponding protein complexes. While rye has been largely unstudied in this regard, closely related species such as barley and wheat have been extensively investigated [20]. Polymeric, insoluble flavonoids, primarily PAs, are found at varying concentrations in the testa of barley [22], wheat [23], and, as demonstrated in our study, rye. PAs (condensed tannins) are oligomers and polymers of flavan-3-ols (catechins) and flavan-3,4-diols (leucoanthocyanidins) [24]. While PAs are colorless in their native state, they can oxidize to form brown or red pigments. In barley, uncolored oligomeric and polymeric PAs are present in high concentrations in yellow grains [22], while red-grained wheat varieties contain oxidized red polymeric PAs [23].

The initial steps of anthocyanin biosynthesis (the phenylpropanoid pathway) occur in the cytoplasm and on the endoplasmic reticulum membrane. Following synthesis, anthocyanins undergo modifications and are transported to the vacuole for storage as anthocyanin vacuolar inclusions (AVIs), where further structural adjustments may occur. Two primary mechanisms facilitate vacuolar transport: membrane transporters and vesicle-mediated trafficking. Each distinct class of tonoplast transporter exhibits selectivity toward differently modified anthocyanins (e.g., acylated, malonylated, or glycosylated forms) and may require glutathione-S-transferase activity [25]. Notably, some transporters localize to the Golgi apparatus, suggesting a potential role in mediating the transport of proanthocyanidins (PAs) to extracellular compartments, such as the cell wall [25].

Rye possesses a brown (pigmented) cellular layer in the caryopsis, which is part of the seed coat [1]. Our study shows that the brown color of this layer changes to red or blue upon treatment with vanillin or DMACA, respectively, which suggests high levels of insoluble PAs. Staining of the testa was observed in all studied accessions except for the anthocyanin-less lines vi3 and vi6. In vi3, this layer was entirely missing detectable PAs, while in vi6, only traces were detected after staining. By analogy with the five anthocyanin-less and proanthocyanidin-free mutations in barley (*ant13*, *ant17*, *ant18*, *ant21*, and *ant22*) [22], it can be hypothesized that the *vi3* and *vi6* mutations in rye affect both anthocyanin and PA biosynthesis. However, no apparent association was observed between the absence of the brown layer in *vi3* mutants and their color coordinates.

The biosynthesis of PAs in small cereals is directly or indirectly associated with premature seed germination—a negative trait acquired during domestication due to breeding for shorter seed dormancy. In red-grained wheat, a reduced tendency for preharvest sprouting has been linked to transcription factors that regulate not only the accumulation of PAs in the testa but also the biosynthesis of abscisic acid (ABA). Dominant alleles at three *Myb* homeoloci—*Tamyb10-A1/R2* on chromosome 3A, *Tamyb10-B1/R3* on chromosome 3B, and *Tamyb10-D1/R1* on chromosome 3D [26]—control the expression of the entire set of structural PA biosynthetic genes, including a gene encoding the transport protein glutathione *S*-transferase [27]. The proteins encoded by these loci can bind to the promoter of the structural gene for 9-cis-epoxycarotenoid dioxygenase (*NCED*), which catalyzes the rate-limiting step in abscisic acid biosynthesis [28].

Interestingly, the perennial wild rye species *Secale montanum* Guss. exhibits light brown grain coloration. Similarly, uniform light brown or even red grain color is characteristic of wild annual rye species such as *S. sylvestre* Host. and *S. vavilovii* Grossh. *S. sylvestre* was once considered in rye breeding as a potential donor of resistance to preharvest sprouting [3]. Feral rye, an annual weed problematic in winter wheat production, exhibits weedy traits such as seed shattering and long seed dormancy [29]. The seeds of feral rye may germinate in situ or remain viable in the soil for up to 10 years. This variation in seed germination may be associated with the PA content in the grain, which varies in color from yellow to brownish-olive. Thus, the genetic material described in rye may be valuable for addressing the issue of preharvest sprouting in grain cereals, a problem exacerbated by global climate change.

### 3.3. Genetic Implications

In barley, the blue color of the aleurone is controlled by a combination of five genes [30]. The molecular functions of three of these genes have been identified: *HvMpc2/HvMYB4H* and *HvMyc2/HvMYC4H* encode transcriptional factors, while *HvF3′5′H/Hv35H* encodes flavonoid 3′,5′-hydroxylase. These genes form a trigenic cluster, *MbHF35*, located on chromosome 4HL [31]. The activity of these genes leads to the synthesis of delphinidin-based anthocyanins, which impart a blue color to the aleurone. In bread wheat, aleurone color genes (loci) were introduced through chromosomal translocations from diploid wheat and *Agropyron* species [32]. One such locus, *Ba1*, located on a 4D translocation, carries orthologs of the barley genes *TaMYC4D*, *TaMYB4D*, and *TaF35H* [31]. In rye, orthologs of the *MbHF35* cluster are found on chromosome 7R [31], consistent with evolutionary translocations between ancestral Triticeae chromosomes 4 and 7. Barley, wheat, and rye are the only grain cereals known to carry functional alleles in this cluster [31].

It was previously shown that green-grained rye contains delphinidin 3-rutinoside [33]. Blue-grained varieties of barley and wheat contain not only delphinidin derivatives but also other anthocyanins [34]. We also previously found [9] the presence of cyanidin rutinoside alongside delphinidin rutinoside in four out of five green-grained rye lines studied. However, differences in the content of these anthocyanins were sufficient to differentiate green and purple rye grains based on their h° color index values. The key enzyme responsible for delphinidin aglycone production is F3′5′H, whereas cyanidin aglycone synthesis relies on F3′H (flavonoid 3′-hydroxylase), and OMT (O-methyltransferase) mediates the conversion of cyanidin to peonidin aglycone. Thus, differences in aglycone production in rye seed coats may arise from variations in either structural genes (e.g., *F3′5′H, F3′H, OMT*) or regulatory MYB/bHLH genes that activate *F3′5′H* in the aleurone or *F3′H* and *OMT* in the pericarp.

Monohybrid segregation for green versus yellow grain color has been established in rye, with the dominant gene *C* controlling the green phenotype [5]. Homozygotes for the recessive allele at the *C* locus exhibit yellow grain. According to modern understanding, this “Mendelian” gene *C* may be one of the three genes comprising the putative rye *MbHF35* cluster. Alternatively, yellow-grained lines may carry recessive allelic or non-allelic mutations in any of these closely linked genes. This creates challenges in allelism testing and segregation analysis for these genes. The most effective approach to determine active and inactive genes in this cluster in yellow-grained lines may involve analyzing the structure and expression of the genes comprising the putative cluster.

The color of wheat and barley grains with anthocyanins in the aleurone is typically described as blue, whereas in rye, it is more frequently described as green or greenish. In our study, the aleurone of green rye grains appeared blue under microscopic examination. It is reasonable to assume that the green color of the rye grain surface results from the superposition of three differently colored layers: blue aleurone, brown testa, and yellow pericarp, consistent with previous findings [3,4]. In contrast, barley and wheat grains without PAs exhibit a blue anthocyanin color in the aleurone, as the uncolored seed coat does not obscure this coloration.

The violet grain color in certain experimental rye accessions is determined by the dominant gene *Vs* (*Violet seed*), which originates from weedy rye [3,5]. The *Vs* gene functions similarly to the purple pericarp genes in wheat (*Pp3*) [35], barley (*Pre2/Ant2*) [36], and rice (*Kala4*) [37], all of which encode bHLH transcription factors. The expression of these genes in the presence of active alleles of other structural and regulatory genes involved in anthocyanin biosynthesis and are responsible for anthocyanin accumulation in the pericarp and other plant tissues. While the molecular function of the *Vs* gene in rye has not been precisely determined, it has been mapped to chromosome 2R under the designation *Ps* (*Purple seed*) [38], in positional orthology with corresponding genes in wheat [39] and barley [36]. The qualitative and quantitative composition of anthocyanins in the colored pericarp of cereals, including rye [9], varies significantly [40]. However, the most common anthocyanidins across all studied cereals (excluding sorghum) are cyanidin and its 3-*O*-methylated derivative, peonidin.

The brown color of the pericarp in some of our rye lines warrants special discussion, particularly in light of the well-studied genetics of grain color in rice and maize. In rice, the pericarp may contain PAs (red rice), anthocyanins (black rice), both pigments [41], or an unidentified brown pigment. The brown pigment in rice forms in homozygotes for the non-active allele of the structural gene *dihydroflavonol 4-reductase* (*Rd*), in the presence of a dominant allele of either of the two bHLH transcription factor genes, *Rc* [42] or *Kala4* [37]. These transcription factors regulate multiple structural genes beyond *Rd* in the anthocyanin (*Kala4*) or PA (*Rc*) biosynthetic pathways. A block in both pathways may lead to the accumulation of identical precursors, such as dihydroflavonols or other intermediate products, resulting in the formation of a brown pigment. Similarly, in maize, brown pigment formation has been observed in the aleurone of lines homozygous for inactive alleles of the *a1* gene, which encodes the same dihydroflavonol 4-reductase, in the presence of corresponding R2R3-MYB and bHLH transcription factors. Brown pigment also forms in the pericarp of maize lines homozygous for a null allele at the *a1* locus, provided they carry appropriate alleles of the R2R3-MYB gene *P1*, which independently controls phlobaphene synthesis [43,44].

We can assume that the unidentified brown pigment that we found in the investigated rye lines with a certain genetic background for the first time may be a product of a mechanism genetically similar to the formation of the brown grain color in maize and rice, as described above. Brown grains in rye are observed in lines with two specific genotypes: those combining the dominant *Vs* gene with homozygosity for one of two anthocyanin-less mutations—*vi1* or *vi2*. The molecular functions of the *Vs*, *vi1*, and *vi2* genes remain unknown. Notably, brown rye grains contain no anthocyanins [9], and their pericarp does not stain with chemical agents specific for PAs. This suggests that the brown pigmentation in our rye lines is genetically and biochemically distinct from the brown grain coloration observed in open-pollinated rye varieties [3,4].

While our results provide a foundation for the practical use of grain coloration in rye breeding, several limitations should be acknowledged. First, the 26 rye lines used in this study, though representative of our genetic collection and covering greater genetic diversity than most studies focused on industrial lines with common ancestry, may not fully capture the complete genetic diversity of rye. Second, while these lines exhibit stable coloration phenotypes, we did not investigate the potential influence of environmental factors on grain coloration, which likely play a role in pigment biosynthesis. Additionally, the molecular genetic mechanisms underlying the observed brown phenotypes remain largely unexplored, and further functional validation of the identified pigments is required. Finally, the study did not address the potential role of co-pigmentation or fully characterize the brown pigment observed in some lines. Future studies should aim to address these limitations by expanding the sample size, exploring environmental effects, and employing advanced analytical techniques.

## 4. Materials and Methods

### 4.1. Plant Material

Rye lines from the Peterhof genetic collection were grown under open field conditions following conventional practices for winter rye cultivation. The open-pollinated variety *Esto* (*vi1*) has been maintained for decades through cross-pollination of a small number of plants and can be considered an inbred line. The remaining lines are self-fertile inbred lines that have been previously used for studies on grain anthocyanins [9].

The study included 26 lines representing five distinct groups:

In anthocyanin-less rye (lines 1–7 in Table 1), anthocyanin pigmentation is absent throughout the entire plant due to homozygosity for recessive spontaneous mutations: *vi1* (accession numbers 1 and 2), *vi2*, *vi3*, *vi4*, *vi5*, and *vi6* (accession numbers 3, 4, 5, 6, and 7, respectively). The grains of these lines appear yellow.

Four lines with anthocyanin pigmentation of vegetative tissues (lines 8, 9, 10, and 11) also exhibit yellow grains. In these lines, the yellow grain color results from homozygosity for recessive alleles of the genes *Vs* and *C*, which control anthocyanin coloration of the pericarp and aleurone, respectively [5].

According to conventional rye genetics [5], the dominant gene *C* is responsible for green grains (lines 16–22), while the dominant gene *Vs* is responsible for violet grains (lines 23–26). Brown grains are characteristic of homozygotes for the dominant allele *Vs* combined with recessive mutations *vi1* (lines 12–14) or *vi2* (line 15).

### 4.2. Color Coordinate Analysis

Grain images were obtained using an Epson Perfection V19 scanner (Seiko Epson Corp., Suwa, Nagano, Japan). The scanner was pre-calibrated with the Munsell Color Checker Mini (X-Rite Inc., Grand Rapids, MI, USA) calibration target, following the protocol described for the GrainScan program used for image analysis [10]. Scanning was performed at a resolution of 300 dpi. For each grain, three color coordinates (L*, a*, and b*) were obtained in the three-dimensional CIELAB color space with the GrainScan v.1.0.140429 [10] software. These coordinates are device-independent and correspond to human visual perception [12].

Based on the a* and b* values, two additional coordinates—c* and h*--are calculated to describe color in the CIE L*c*h* color space. The vertical L* axis (lightness) remains the same, ranging from 0 to 100. c* (chroma, or color saturation) is calculated as (a*^2^ + b*^2^)^^1/2^. h* (hue, or actual color) is calculated in degrees (h°) or radians as the arctangent of b*/a*. On average, approximately 100 grains were individually scanned for each of the 26 samples. The exact numbers are provided in Table 1. The mean values of the coordinates for each color group were compared using ANOVA in PAST v.4.03 [45]. Significant differences between means were determined using Tukey’s range test at *p* < 0.05 [11].

### 4.3. Histological Procedures and Microscopy of Pigment Distribution

Grains at the soft dough stage (17–20 days after pollination, depending on the sample) were frozen in OCT compound (Sakura Finetek USA Inc., Torrance, CA, USA) and sectioned into 25 mkm slices using a Leica CM-3050S cryomicrotome (Leica Biosystems, Nussloch, Germany) at −20 °C.

For anthocyanin localization, slices were placed directly onto cold slides and mounted under a coverslip with cold (+6 °C) 1-hexadecene (Sigma-Aldrich, St. Louis, MO, USA) to prevent the delocalization of water-soluble anthocyanins at this developmental stage. The coverslip was affixed and sealed with a nail polish.

For proanthocyanidin (PA) localization, slices were thaw-mounted onto poly-L-lysine-coated slides and allowed to dry at +37 °C for 2 h. Two PA-specific staining methods were employed:Vanillin-HCl Staining: Slices were incubated in absolute ethanol containing 50 mg/mL vanillin (Sigma-Aldrich, St. Louis, MO, USA) for 30 s, followed by the immediate application of a drop of concentrated HCl [46]. The slices were then coverslipped and immediately examined under a microscope.DMACA Staining: Slices were incubated for 20 min in a freshly prepared solution of 0.01% (*w*/*v*) 4-dimethylaminocinnamaldehyde (Sigma-Aldrich, St. Louis, MO, USA) and 0.8% (*v*/*v*) concentrated HCl in absolute ethanol [47]. After incubation, the slices were washed three times with 70% ethanol and coverslipped using self prepared Hoyer’s medium according to Cold Spring Harbour protocol [48].

Microscopy was performed using a Leica DM5500 microscope (Leica Microsystems, Wetzlar, Germany) controlled by MicroManager v2.0 software [49]. Images were acquired using 20× (0.7 NA) and 40× (0.85 NA) objectives. Montages of multiple fields of view, covering the entire slice, were automatically captured using MicroManager v2.0, and panoramic stitching was performed using FiJi v.1.53h [50].

### 4.4. MALDI-Imaging of Anthocyanins Localization

Grains at the soft dough stage (17–20 days after pollination) were cryosectioned according to the Kawamoto technique [51]. Briefly, grains were immersed in 0.5% carboxymethylcellulose (Sigma-Aldrich, St. Louis, MO, USA) and rapidly frozen in *n*-hexane at −80 °C. The frozen blocks were mounted on holders and sectioned into 12 mkm slices using a Leica CM-3050S cryomicrotome (Leica Microsystems, Wetzlar, Germany) at −20 °C. The slices were captured on Cryofilm type 2C (Section-Lab, Yokohama, Japan) and freeze-dried in the cryomicrotome’s chamber for 4 h to prevent the diffusion and delocalization of water-soluble anthocyanins. Slices on Cryofilm were affixed to ITO-coated electrically conductive slides (Bruker, Bremen, Germany) using conductive carbon tape (16073-2, Ted Pella, Redding, CA, USA). Fiducial points were marked with a white marker (Edding 750, Ahrensburg, Germany), and optical images of the slides were acquired using a Bio-Rad GS-800 scanner (Bio-Rad, Hercules, CA, USA) at 700 dpi. As an external control and calibration standard, bilberry anthocyanin extract (Y0001059, bilberry dry extract, Sigma-Aldrich, St. Louis, MO, USA) was spotted near the fiducial points. The slices were then coated using an automated ImagePrep coater (Bruker Daltonics GmbH & Co. KG, Bremen, Germany, version 2.0.1) with the built-in “DHB_nsh04” protocol. The matrix, 2,5-dihydroxybenzoic acid (DHB; 40 mg/mL, Bruker Daltonics GmbH & Co. KG, Bremen, Germany), was dissolved in a 70:30 (*v*/*v*) methanol/deionized water solution containing 0.1% trifluoroacetic acid (TFA; Sigma-Aldrich, St. Louis, MO, USA).

Mass spectra were acquired using a MALDI-TOF Ultraflextreme instrument (Bruker Daltonics GmbH & Co. KG, Bremen, Germany) controlled by Bruker FlexControl v. 3.3 and FlexImaging v. 3.0 software. The laser intensity was set to 60%, with a “minimal” laser spot size and a raster size of 50 mkm. Each raster point was sampled with 250 laser shots, and a random walk pattern was applied within each point. The actual laser spot size, measured after matrix ablation, was approximately 20 mkm. Data acquisition was performed in high-resolution reflectron mode for positive ions, with an extraction delay of 80 ns and a mass range of 230–1200 Da. For MS/MS, the isolation window was set to ±2 Da, and laser-induced dissociation with 100% laser intensity was used for fragmentation.

Optical images were co-registered in FlexImaging by aligning the fiducial points. The data were converted from Bruker’s proprietary format to imzML format using the proteowizard’s “msconvert.exe” utility [52] and imzMLConverter_1.3 [53]. Visualization and data analysis were performed using the Cardinal package for R v2.8 [54]. The data were normalized by total ion current, smoothed using a Savitzky-Golay filter with a 10 Da window, baseline-corrected using a median filter with a 100 Da block size, and subjected to peak detection using the Limpic algorithm.

## 5. Conclusions

We demonstrated that the GrainScan software method [10] is suitable for the objective measurement of rye grain color. The four main visually identified color groups—lines with yellow, green, brown, and violet grains—were effectively differentiated based on the value of the color coordinate h°. These color coordinates can be used directly or incorporated into custom indices for sophisticated analyses of intra-group color variation, as well as for non-destructive seed analysis and sorting to support breeding efforts.

Our findings confirmed distinct pigment localization in green and violet grains, with different sets of anthocyanins identified in the aleurone (delphinidin glycosides) and pericarp (peonidin glycosides), as supported by MALDI-imaging analysis of selected lines. Additionally, our analysis of PAs presence in anthocyanin-less lines (*vi1*–*vi6*) revealed their varying concentrations in the testa: high concentrations in *vi2* and *vi5*, low concentrations in *vi1* and *vi6*, and a complete absence of PAs in *vi3*.

These results detailed the phenotypic results of expression of non-allelic genes *vi1*–*vi6* and *Vs* at the anatomical level. Subsequent studies should be aimed at elucidating the molecular function of these genes and analyzing their influence on the composition of flavonoids in rye grain separately and in different combinations. That will offer a foundation and trait donors for developing specialized rye varieties that either completely lack PAs and anthocyanins or exhibit high concentrations of specific anthocyanins in different seed layers.

## Figures and Tables

**Figure 1 plants-14-02557-f001:**
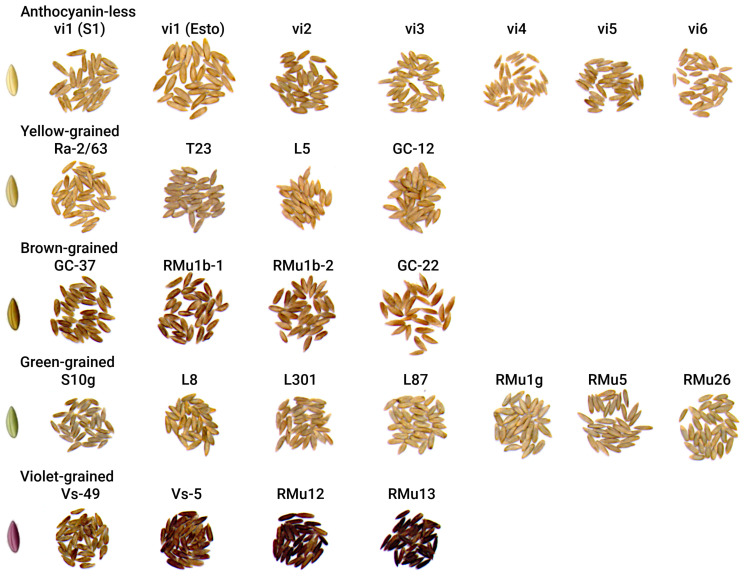
Diversity of rye grain coloration.

**Figure 2 plants-14-02557-f002:**
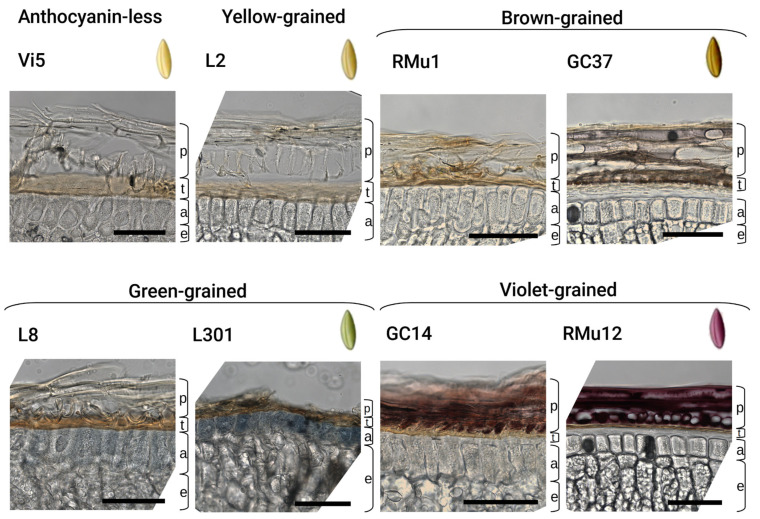
Pigment localization in grains of different colors, p-pericarp, t-testa, a-aleurone, e-endosperm. Scale bar is equal to 100 mkm.

**Figure 3 plants-14-02557-f003:**
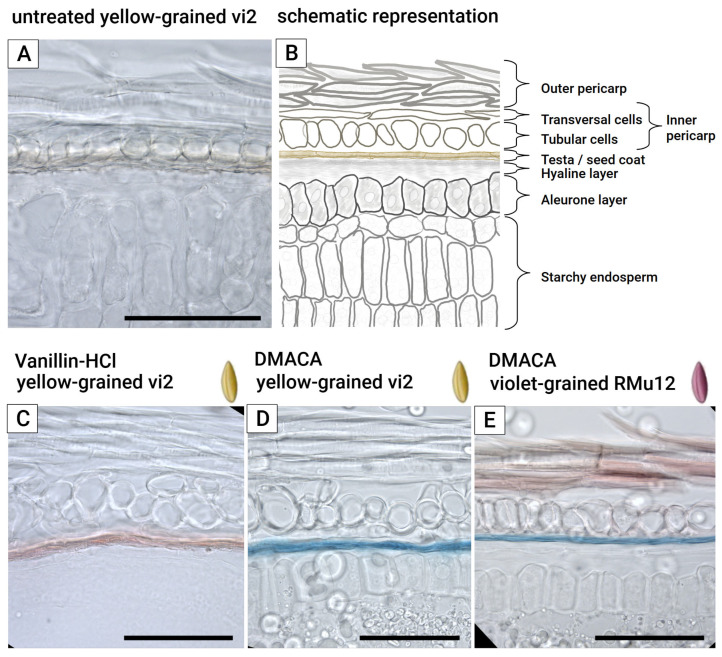
Vanillin-HCl and DMACA staining of PAs in slices of rye grains. (**A**) Untreated slice of yellow grain, (**B**) schematic representation of (**A**), (**C**) Vanillin-HCl treatment of yellow grain, (**D**) DMACA staining of yellow grain, (**E**) DMACA staining of violet grain. Scale bar is equal to 100 mkm.

**Figure 4 plants-14-02557-f004:**
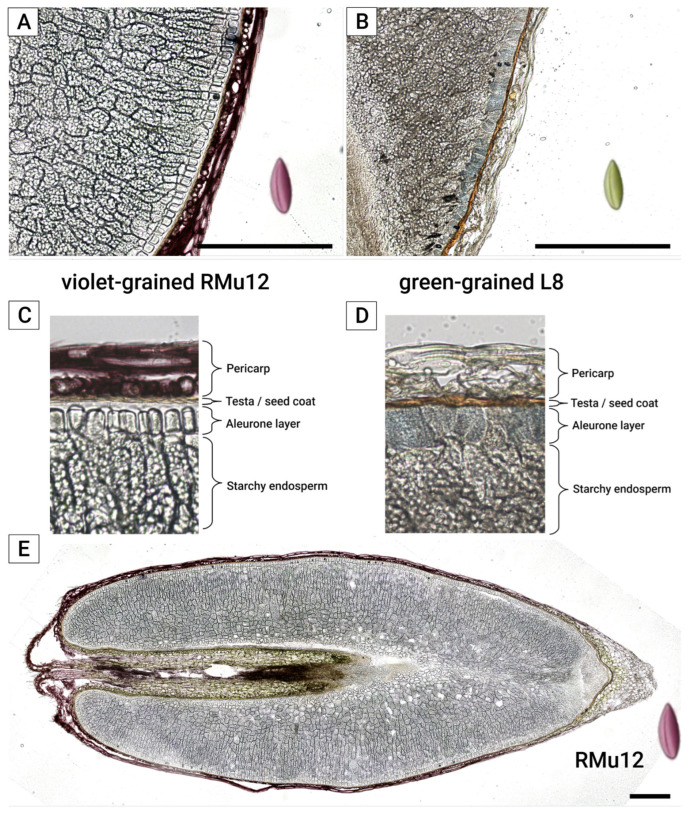
Anthocyanin distribution on histological slices of violet and green grains of rye. (**A**,**C**) Violet-grained line RMu12, (**B**,**D**) green-grained line L8, (**E**) whole grain of line RMu12. Scale bar is equal to 500 mkm.

**Figure 5 plants-14-02557-f005:**
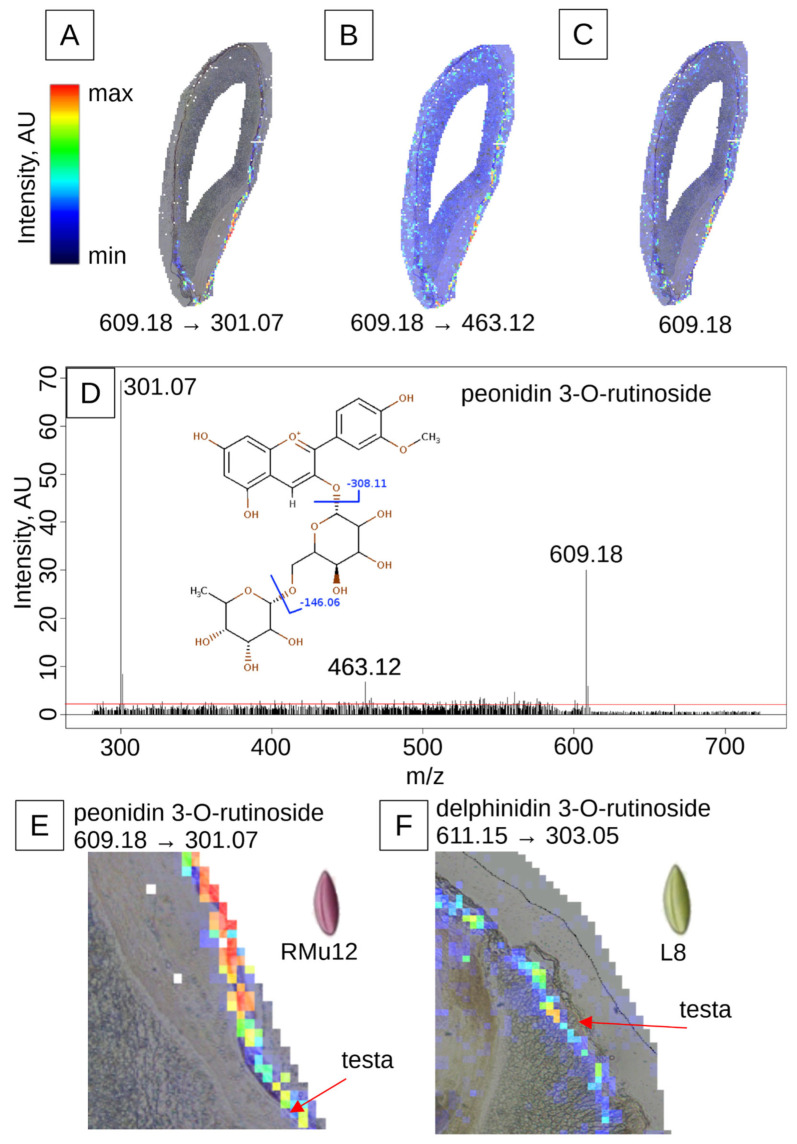
MALDI-IMS analysis of peonidin 3-*O*-rutinoside and delphinidin 3-*O*-rutinoside distribution. (**A**–**C**) Violet-grained rye line RMu12, co-registered with a histological slice: (**A**) aglycone fragment *m*/*z* 301.07 (−308.11 Da), (**B**) fragment *m*/*z* 463.12 (−146.06 Da), (**C**) molecular ion *m*/*z* 609.18, color scale: black (absence), blue (minimal intensity), green (intermediate intensity), red (maximal intensity). (**D**) Fragmentation pattern of the molecular ion: (**E**) distribution of peonidin 3-*O*-rutinoside in the pericarp of violet-grained rye, as evidenced by the intensity of the aglycone fragment (*m*/*z* 301.07), maximal intensity is observed superficially to the testa, denoted by a red arrow, (**F**) distribution of delphinidin 3-*O*-rutinoside in the aleurone of green-grained line L8, as evidenced by the intensity of the aglycone fragment (*m*/*z* 303.05); maximal intensity is observed below the testa, which is denoted by a red arrow.

**Figure 6 plants-14-02557-f006:**
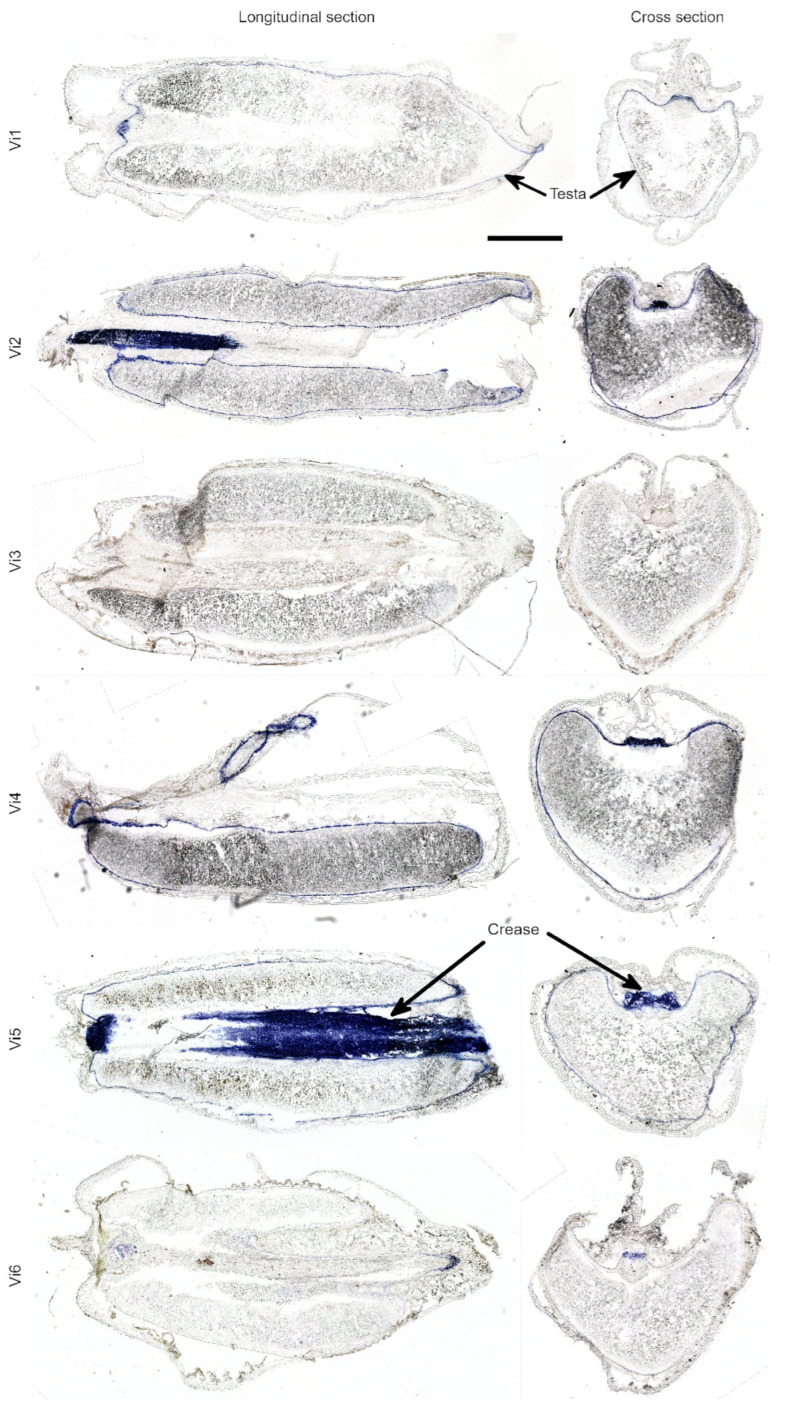
DMACA staining of longitudinal and cross sections of grains in anthocyanin-less mutant lines (**vi1**–**vi6**). Dark blue staining indicates the presence of PAs in the testa and crease, as shown by arrows. Scale bar: 1000 mkm.

**Table 1 plants-14-02557-t001:** CIELAB color coordinates of grains in rye lines.

Accessions ^1^	N	L*	a*	b*	C*	h°	Color ^3^
**Anthocyanin-less:**							
1. vi1 (S1)	90	49.09 ± 2.12	3.69 ± 0.54	15.19 ± 0.60	15.64 ± 0.64	76.36 ± 1.83	
2. vi1 (Esto)	98	48.24 ± 2.23	3.83 ± 0.70	15.56 ± 0.82	16.04 ± 0.90	76.23 ± 2.08	
3. vi2	96	45.74 ± 1.96	3.86 ± 0.75	15.46 ± 0.64	15.95 ± 0.73	76.06 ± 2.39	
4. vi3	94	50.14 ± 1.75	3.19 ± 0.48	17.27 ± 0.68	17.56 ± 0.72	79.56 ± 1.39	
5. vi4	94	49.24 ± 2.54	3.06 ± 0.38	14.80 ± 0.52	15.11 ± 0.56	78.35 ± 1.19	
6. vi5	91	48.66 ± 3.18	3.81 ± 0.76	16.16 ± 1.09	16.62 ± 1.18	76.80 ± 2.05	
7. vi6	100	50.73 ± 2.11	3.29 ± 0.83	15.66 ± 1.04	16.02 ± 1.10	78.18 ± 2.64	
**Mean 1–7**		**48.83 a ^2^**	**3.53 b**	**15.73 a**	**16.13 a**	**77.36 b**	
**Yellow:**							
8. Ra-2/63	103	45.10 ± 2.55	3.02 ± 0.57	14.66 ± 1.17	14.98 ± 1.19	78.36 ± 2.03	
9. T23	93	46.76 ± 1.95	2.93 ± 0.45	13.35 ± 0.85	13.67 ± 0.90	77.64 ± 1,48	
10. L5	106	46.07 ± 2.22	3.82 ± 0.91	14.74 ± 0.96	15.25 ± 1.05	75.57 ± 3.04	
11. GC-12	100	45.04 ± 2.53	3.06 ± 0.50	14.69 ± 1.16	15.01 ± 1.17	78.22 ± 1.76	
**Mean 8–11**		**45.74 a**	**3.21 b**	**14.36 ab**	**14.73 ab**	**77.45 b**	
**Brown:**							
12. GC-37	90	39.80 ± 4.06	4.70 ± 0.78	14.65 ± 0.82	15.40 ± 0.85	72.24 ± 2.78	
13. RMu1b-1	89	44.44 ± 2.79	4.10 ± 0.62	14.87 ± 0.13	15.43 ± 1.20	74.53 ± 2.40	
14. RMu1b-2	135	38.93 ± 3.47	4.80 ± 0.69	15.40 ± 0.99	16.14 ± 1.08	72.74 ± 1.94	
15. GC-22b	93	36.81 ± 3.15	5.05 ± 0.80	15.72 ± 0.83	16.53 ± 0.84	72.22 ± 2.44	
**Mean 12–15**		**40.00 b**	**4.66 a**	**15.16 ab**	**15.88 ab**	**72.81 c**	
**Green:**							
16. S10g	100	48.60 ± 2.27	2.18 ± 0.66	13.10 ± 0.93	13.29 ± 0.98	80.65 ± 2.52	
17. L8	100	43.99 ± 2.48	2.74 ± 0.74	15.13 ± 1.02	15.38 ± 1.24	79.87 ± 4.65	
18. L301	102	47.07 ± 2.05	1.96 ± 0.42	13.51 ± 0.70	13.66 ± 0.73	81.77 ± 1.55	
19. L87	102	49.46 ± 2.32	2.02 ± 0.63	14.86 ± 0.99	15.00 ± 1.06	82.36 ± 1.95	
20. RMu1g	109	49.06 ± 2.32	1.87 ± 0.51	12.65 ± 1.04	12.80 ± 1.06	81.64 ± 2.05	
21. RMu5	133	46.43 ± 2.09	2.27 ± 0.62	13.03 ± 0.94	13.24 ± 1.01	80.24 ± 2.13	
22. RMu26	101	45.14 ± 2.68	1.91 ± 0.87	14.28 ± 1.32	14.43 ± 1.40	82.58 ± 2.98	
**Mean 16–22**		**47.11 a**	**2.14 c**	**13.79 b**	**13.97 bc**	**81.30 a**	
**Violet:**							
23. Vs-49	64	33.53 ± 3.52	5.05 ± 0.94	13.31 ± 1.06	14.26 ± 1.11	69.23 ± 3.52	
24. Vs-5	92	32.19 ± 2.85	6.01 ± 0.76	12.11 ± 1.14	13.54 ± 1.28	63.54 ± 3.3	
25. RMu12	91	24.06 ± 2.36	3.02 ± 0.97	8.24 ± 1.00	8.80 ± 1.22	70.36 ± 4.69	
26. RMu13	83	25.02 ± 3.44	4.61 ± 1.20	9.06 ± 1.55	10.20 ± 1.81	63.30 ± 4.33	
**Mean 23–26**		**28.70 c**	**4.67 a**	**10.68 c**	**11.71 c**	**66.61 d**	

^1^ 1–7 anthocyanin-less lines, 8–11 yellow-grained lines with anthocyanin pigmentation in vegetative organs, 12–15 brown-grained lines, 16–22 green-grained lines, 23–26 violet-grained lines. The values for coordinates are the mean value ± standard deviation; ^2^ Mean values marked by the same letter in each column are not significantly different at *p* ≤ 0.05; ^3^ Mean perceived color of rye grains of each accession and the mean color for each group of accessions.

**Table 2 plants-14-02557-t002:** Histological analysis of pigment and PAs localization in grains of selected rye lines.

PAs Testa	PAs Crease	Aleurone	Testa	Pericarp	Accessions
**Anthocyanin-less:**					
blue	blue	uncolored	light brown	uncolored	1. vi1 (S1)
uncolored	slight blue	uncolored	brown	uncolored	2. vi1 (Esto)
blue	dark blue	uncolored	light brown	uncolored	3. vi2
uncolored	uncolored	light greenish	light greenish	uncolored	4. vi3
light blue	slight blue	uncolored	brown	uncolored	5. vi4
light blue	blue	uncolored	light brown	uncolored	6. vi5
uncolored	very slight blue	uncolored	slight yellow	uncolored	7. vi6
**Yellow:**					
blue	blue	uncolored	yellow	uncolored	8. Ra-2/63
light blue	blue	yellowish	dark brown	uncolored	9. T23
light blue	blue	uncolored	brown	uncolored	10. L5
light blue	blue	uncolored	brown	uncolored	11. GC-12
**Brown:**					
blue	blue	uncolored	dark brown	brown	12. GC-37
blue	blue	uncolored	light brown	light brown	13. RMu1b-1
blue	blue	uncolored	brown	light brown	14. RMu1b-2
brown	blue	uncolored	brown	brown	15. GC-22b
**Green:**					
dark blue	dark blue	very slightly blue	brown	uncolored	16. S10g
slight blue	slight blue	very slightly blue/blue	light brown	uncolored	17. L8
slight blue	slight blue	blue	brown	uncolored	18. L301
dark blue, blue aleurone	dark blue	blue	dark brown	uncolored	19. L87
very slight blue	slight blue	very slightly blue	brown	uncolored	20. RMu1g
uncolored	uncolored	light-blue	uncolored	uncolored	21. RMu5
uncolored	uncolored	blue	brown	uncolored	22. RMu26
**Violet:**					
red	slight blue	uncolored	brown	light brown	23. V-49
slight blue	blue	uncolored	brown	brown-violet	24. V-5
dark blue	blue	uncolored	dark brown	dark violet	25. RMu12
slight brown	slight blue	very slightly pink	brown	violet	26. RMu13

## Data Availability

The original contributions on grain color presented in this study are included in the article. The raw data supporting the conclusions of this article will be made available by the authors on request. Further inquiries can be directed to the corresponding author.

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
