# Peer review of "Anatomical and Digital Image Analysis of Flavonoid-Mediated Grain Coloration in Rye (Secale cereale L.)"

_plants, 2025, doi:10.3390/plants14162557_

Round 1
Reviewer 1 Report
Comments and Suggestions for Authors
I have reviewed the manuscript titled “Anatomical and digital image analysis of flavonoid-mediated grain coloration in rye (Secale cereale L.)”. The subject is interesting and presents some novelty. The paper is well presented and easy to read. The literature cited is relevant to the study. I think the paper could prove to be very interesting and useful to very large researchers, possibly making it acceptable for publication in Plants journal after minor revisions.
I would like to give further suggestion on the following matters:
Line 132: Statisrical evaluation should be added to each data.
Line 211: The resolution of Fig. 6 should be enhanced.
Line 671: Add the editor names of the book.
Author Response
Comment 1: I would like to give further suggestion on the following matters:
Line 132: Statisrical evaluation should be added to each data.
Response 1: The statistical evaluation of the data provided in the Table 1 is described in material and methods, lines 568-573. The Table 1 contain values of the coordinates ± SD (standard deviation). We've improved the table description to reflect this, lines 134-135.
Comment 2: Line 211: The resolution of Fig. 6 should be enhanced.
Response 2: We agree to this point, the pdf version of the manuscript, unfortunately, contained a downsized version of the Fig.6 to keep the pdf file size reasonably small. The original image uploaded separately for the publication is 4516x2479 pixels (19 Mb in lossless PNG format). Its resolution allows for precise location of blue stain in respect to seed coat layers and allows to see separate cells in the aleurone layer.
Comment 3: Line 671: Add the editor names of the book.
Response 3: Thank you! The citation is corrected, reflecting the specific chapter of the book:
-
Abayomi, O.O.; Gan, C.-Y.; Shafie, M.H.; Alenezi, H.; Taiwo, A.E.; Olumide, F.S. Nutritional quality of color cereals and effects of processing on its functional properties. In Functionality and Application of Colored Cereals Nutritional, Bioactive, and Health Aspects; Academic Press: Cambridge, MA, USA, 2023; pp. 27–46. doi:10.1016/b978-0-323-99733-1.00003-0
Reviewer 2 Report
Comments and Suggestions for Authors In this manuscript (plants-3797977), the authors investigated the relationships between grain color and flavonoid distribution using the rye grains in 5 different color groups by colorimetrical, histochemical, and mass spectrometry methods. Presented data are obvious and visually understandable. However, the discussion section is too lengthy, particularly in “Genetic Implications”, without referring data in this manuscript. Each comment was described bellow.
- Table 1 In a* and h values, the letters showing significant difference weren’t used in alphabetical order. “a” should be used on the largest value in column.
- Lines 220–223: In referenced report [14], the a* positively correlated with total anthocyanin content, so this sentence isn't correct..
- Lines 245-247: This sentence contradicted with data shown in Table 1 and Figure 2. The a* positively correlate with TAC, didn't it.
Author Response
Comment 1: However, the discussion section is too lengthy, particularly in “Genetic Implications”, without referring data in this manuscript. Each comment was described bellow.
Response 1: Thank you for your insightful comment. We would like to clarify why discussing the genetic implications is important.
The findings presented in this manuscript are a part of a broader study of anthocyanin synthesis and regulation mechanisms in rye (ref. 5–9), an area where knowledge remains limited. Our results demonstrate that: blue pigment accumulates exclusively in the aleurone layer of green-grained lines; brown pigment is found in the pericarp and testa of brown-grained lines. Notably, the chemical nature of the brown pigment remains unknown, and the particularities of biosynthetic pathways for both brown and green pigments in rye are yet to be elucidated.
In the "Genetic Implications" section, we discuss the genetic control of green and brown pigment synthesis in other Poaceae species, which provides a foundation for future research on rye. The histological findings are valuable not only on their own but also for their potential integration into subsequent genetic and molecular studies.
Comment 2:
-
Table 1 In a* and h values, the letters showing significant difference weren’t used in alphabetical order. “a” should be used on the largest value in column.
Response 2: Thank you for pointing this out ! We've updated the table accordingly (Line 132).
-
Comment 3:
-
Lines 220–223: In referenced report [14], the a* positively correlated with total anthocyanin content, so this sentence isn't correct.
Response 3: In the article, the exact sentence citing Ref. 14 is as follows (lines 222-225):
”For example, in colored varieties of grape (Vitis vinifera L.), all color coordinates (L*, a*, b*, and C*) exhibit a negative correlation with total anthocyanin content (TAC), while h° shows a positive correlation [14].”
In Ref. 14, a range of different indices was calculated to assess the correlation between total anthocyanin content and color coordinates (L, a, b, C*, and h°). Among them, the CIRG2 and CIRWG (Color Index for Red Wine Grapes) showed the best potential for evaluating the color of wine cultivars (Table 2). These indices were calculated as: CIRG2 = (180 - H) / (L* × C) and CIRWG = [h / (L* × b)] × 100. A higher anthocyanin content in wine berries corresponds to higher values of these indices. Based on the equations, h° exhibited higher values, while L, a, b, and C had lower values. This indicates that the color coordinates (L, a, b, and C) are negatively correlated with total anthocyanin content (TAC), whereas h° shows a positive correlation - consistent with the findings mentioned in our article (Lines 222-225).
-
Comment 4:
-
Lines 245-247: This sentence contradicted with data shown in Table 1 and Figure 2. The a* positively correlate with TAC, didn't it.
Response 4: Thank you very much for your comment. We agree with that and revised the sentence.
In the article, the exact sentence is as follows (lines 246-248): Considering previous study on rye [9] and studies on wheat [17] and grape [13], we identified similar relationships between anthocyanin concentration and color coordinates. Specifically, L*, a*, b*, and C* exhibited negative correlations with TAC, while h° showed a positive correlation.
The revised sentence (Lines 247-255):
Based on previous studies of rye [9], wheat [17], and grape [13], we observed similar relationships between anthocyanin concentration and color coordinates (Table 1). Specifically, L, b, and C* displayed negative correlations with total anthocyanin content (TAC). However, unlike the findings in [9, 13, 17], h° also exhibited a negative correlation with TAC in our study. Additionally, the a* coordinate for lines with the highest TAC (violet-colored lines) was similar to that of brown-colored lines lacking anthocyanins in grains [9], indicating that a* is not a reliable indicator for assessing anthocyanin content in rye.
Reviewer 3 Report
Comments and Suggestions for Authors
Review of an article by Pavel A. Zykin and co-authors. The article concerns the occurrence of anthocyanins in different varieties of Secale cereale. The authors skillfully presented the research problem and the role of the study. I must commend the authors for using sophisticated techniques for histology and showing pigment distribution. The documentation is very good. The authors solved the research problem and skillfully discussed their results with the literature data.
I consider the article to be very good and recommend its publication. I have a minor suggestion.The authors should add to the discussion in which cellular compartments the pigments are stored. In addition, it would be good if they added a few sentences about which organelles participate in the biosynthesis of anthocyanins.
Author Response
Comment 1: The authors should add to the discussion in which cellular compartments the pigments are stored. In addition, it would be good if they added a few sentences about which organelles participate in the biosynthesis of anthocyanins.
Response 1: Thank you! We've added the information on the cellular compartments involved in the pigments storage and the organelles, participating in the biosynthesis of anthocyanins, lines 349-362, and added the reference #25.